# Utilisation of private general practitioners to provide caesarean deliveries in five rural district public hospitals in South Africa: a mixed-methods study

Tanya Doherty [ID],[1,2,3] Geetesh Solanki [ID],[1,4,5] Emmanuelle Daviaud,[1] Yvandi Bartmann,[1] Anthony Hawkridge,[6] Sue Fawcus[7]

**Correspondence to**
Prof Tanya Doherty;
tanya.doherty@mrc.ac.za

## ABSTRACT

**Objective** Researching how public–private engagements may promote universal access to safe obstetric care including caesarean delivery is essential. The aim of this research was to document the utilisation of private general practitioners (GPs) contracted to provide caesarean delivery services in five rural district hospitals in the Western Cape, the profile and outcomes of caesarean deliveries. We also describe stakeholder experiences of these arrangements in order to inform potential models of public–private contracting for obstetric services.

**Design** We used a mixed-methods study design to describe rural district hospitals' utilisation of private GP contracting for caesarean deliveries. Between April 2021 and March 2022, we collated routine data from delivery and theatre registers to capture the profile of deliveries and maternal outcomes. We conducted 23 semistructured qualitative interviews with district managers, hospital-employed doctors and private GPs to explore their experiences of the contracting arrangements.

**Setting** The study was conducted in five rural district hospitals in the Western Cape province, South Africa.

**Results** The use of private GPs as surgeon or anaesthetist for caesarean deliveries differed widely across the hospitals. Overall, the utilisation of private GPs for anaesthetics was similar (29% of all caesarean deliveries) to the utilisation of private GPs as surgeons (33% of all caesarean deliveries). The proportion of caesarean deliveries undertaken by private GPs as the primary surgeon was inversely related to size of hospital and mean monthly deliveries. Adverse outcomes following a caesarean delivery were rare. Qualitative data provided insights into contributions made by private GPs and the contracting models, which did not incentivise overservicing.

**Conclusion** The findings of this study suggest that private GPs can play an important role in filling gaps and expanding quality care in rural public facilities that have insufficient obstetric skills and expertise. Different approaches to enable access to safe caesarean delivery are needed for different contexts, and contracting with experienced private GP's is one resource for rural district hospitals to consider.

## STRENGTHS AND LIMITATIONS OF THIS STUDY

⇒ This is one of the first studies to describe utilisation of private general practitioners to support caesarean deliveries within rural district hospital settings in a low/middle-income country context.

⇒ A mixed-methods study design including routine obstetric data and qualitative interviews provides a robust description of experiences with this public–private model.

⇒ We collated obstetric data from registers and did not perform individual patient folder reviews. The proportion of caesarean delivery complications could have been higher if individual folders were reviewed.

⇒ We did not collect data on neonatal outcomes, which is also an important indicator of the quality and safety of caesarean delivery.

## INTRODUCTION

Maternal and child health is a major public health concern in South Africa (SA). While there has been progress in reducing maternal mortality to 119 per 100 000 live births,[1] there remain huge disparities between public and private sectors in the distribution of obstetric care providers, models of care and outcomes.[2] The caesarean delivery rate in the public sector is 28%[2] compared with 74% in the private sector.[3] A global analysis exploring the relationship between health financing indicators and caesarean delivery rates found that in 2018, there were a total of 8.8 million unnecessary caesarean deliveries globally, two-thirds of which occurred in upper middle-income countries.[4] The study reported that a 10% increase in voluntary health insurance was associated with a 4% increase in excess caesarean deliveries, defined as caesarean delivery proportions above the global target of 19%.[4]

In line with patterns seen in other low/middle-income countries (LMICs),[5] the case fatality rate (CFR) for caesarean deliveries in the SA public sector is three times higher than for vaginal delivery, and 27% of the caesarean delivery-related CFR is associated

with haemorrhage.[2] Caesarean delivery CFR and haemorrhage rates are not routinely measured in the private sector so such data are not available. The 2014–2016 Saving Mothers report for SA described a review of maternal deaths in the private sector. The maternal mortality rate was 45 per 100 000 live births (one-third that of the national SA rate at the time) with a similar proportion of deaths from obstetric haemorrhage (19%), of which 17% were from caesarean delivery-associated bleeding.[6] Improved clinical governance and skills development in the public sector are critical to reducing this disparity, but these efforts could be complemented by the engagement of health professionals with obstetric skills in the private sector.

In efforts towards achieving Universal Health Coverage (UHC), SA is in the process of developing policy to allow implementation of a National Health Insurance (NHI). The NHI Bill[7] proposes that NHI will be a strategic purchaser of healthcare services for the entire SA population. The proposed NHI will contract for obstetric services from both public and private providers. A key challenge in SA regardless of when and in what form NHI is implemented is to develop contracting models that allow public and private resources to be mobilised to the benefit of the entire population. This is particularly important for rural district hospitals where surgical skills and supervision may be insufficient to meet the service requirements for obstetric care, and specialist services are not readily available in the public or private sectors.

However, attempts to harness the resources of the private sector to service the needs of the broader SA population would need to address the challenge of ensuring that the care provided is appropriate and patterns of inappropriate care, including the high caesarean delivery rate of the private sector,[3] are not reproduced for the entire population. The development of contracting models that would best mitigate these risks is critical in delivering UHC.

Researching and documenting how private sector resources can be harnessed to support the public health system are urgently needed now, in the transition, and full implementation phases of NHI in SA and also to inform developments in other African countries that are implementing similar NHI financing arrangements. These include countries such as Ghana, Kenya, Nigeria and Tanzania.[8] There are some lessons to be learnt from public–private contracting for obstetric services in India where states have implemented different models of contracting arrangements to increase access to obstetric care.[9] In some states, private obstetricians are contracted in to public facilities to increase skills and human resources, while in other states, costs for services are subsidised within private facilities.[10]

Research to understand whether public–private engagements could reduce inequities and promote universal access to safe obstetric care including caesarean delivery is essential. Understanding the nature of public–private engagements in the area of obstetric care is

particularly important because caesarean delivery is the most commonly performed surgical procedure and the findings may also be useful in informing strategies for other medical and surgical disciplines where similar patterns of underservicing and overservicing in the public versus private sector may occur.

We carried out research to document the utilisation of private general practitioners (GPs) contracted to provide caesarean delivery services in five rural district hospitals in the Western Cape and to qualitatively describe the experiences and perspectives of managers and doctors involved with a view to informing the development of public–private arrangements for improving obstetric and maternal outcomes in rural areas in South Africa and LMICs more broadly. The aim of this paper is to document our findings on the utilisation of the GPs, the profile and outcomes of caesarean deliveries. We also describe stakeholder experiences of these arrangements in order to inform the development of models of public–private contracting for obstetric services.

## METHODS
### Study design
We undertook descriptive health systems research using a mixed-methods study design to describe rural district hospitals' utilisation of private GP contracting for caesarean deliveries in the Western Cape province and the profile and maternal outcomes of these deliveries in five hospitals. We chose a mixed-methods study design incorporating both quantitative and qualitative data collection as this approach is well suited to applied health systems research.[11 12]

### Study setting
The setting for this research was the Western Cape province where existing public–private contracting for caesarean delivery services was occurring due to human resource shortages in rural district hospitals. Five rural district hospitals within one rural district were chosen following engagement with provincial managers and obstetric clinical managers.

In SA, women with low-risk pregnancies receive antenatal care at primary care clinics and community health centres. District hospitals provide level 1 (generalist) services to inpatients and outpatients including obstetric care for women with low-risk pregnancies. District hospitals have between 30 and 200 beds, a 24-hour emergency service and an operating theatre. Generalists (medical officers) provide the services together with nursing staff and allied health professionals; some district hospitals have specialist family physicians serving as clinical managers but there are no obstetric or anaesthetic specialists at district hospital level. Most district hospitals also have community service doctors. These are doctors who have completed a 2-year internship and are required to complete a further 1 year of community service.[13] None of the five hospitals had newly qualified intern medical

doctors who are generally not placed within district hospitals.

For obstetric services at district hospital level, normal vaginal deliveries are performed by midwives, assisted vaginal deliveries are performed by advanced midwives or medical officers and caesarean deliveries (surgery and anaesthesia) are performed by medical officers. Pregnant women with pre-existing morbidities such as diabetes, autoimmune disorders, thyroid disease, and cardiac disease or obstetric complications such as anticipated preterm delivery, suspected intrauterine growth restrictions, pre-eclampsia, placenta praevia, abruptio placentae, multiple pregnancy, two previous caesarean sections, body mass index over 35–40 kg/m², and severe anaemia are referred for delivery to a secondary or tertiary level hospital.

Public health facilities are permitted to contract the services of private providers where needed. There are three mechanisms by which private providers can be contracted to the public service: through a locum agency, through a sessional contract which is limited to a maximum of 39 hours per month or as a service provider in response to a tender for specific services. In all three contracting models, the remuneration is time based and not related to the number of patients or theatre cases performed. In the case of obstetric services, private providers are mainly used for theatre services either as a GP surgeon or GP anaesthetist to undertake caesarean deliveries or for obstetric surgery including ectopic pregnancy, termination of pregnancy and dilatation and curettage following spontaneous miscarriage. They may also be called for an assisted delivery if the establishment doctor is unable to manage a complicated delivery. For GPs contracted through a sessional contract, medicolegal indemnity is provided by the state but for those contracted as locums or through a service provider tender, they are required to have their own medicolegal indemnity cover. In these five hospitals, the private GPs did not have medical indemnity for private obstetric practice and only performed caesarean deliveries during their public sector contracted time.

### Quantitative data collection

For a period of 12 months (1 April 2021–31 March 2022), we collated quantitative obstetric clinical data from all five participating hospitals. Clinical outcomes were captured from the delivery register, theatre register and obstetric transfer book. Outcomes that could be collated from these sources without individual patient folder review included: mode of delivery and caesarean delivery complications (maternal death, postpartum haemorrhage (PPH) and referral to a regional hospital). Other procedures performed at or after a caesarean delivery were also captured as a measure of complications (hysterectomy, B-lynch suture and re-look laparotomy). For each caesarean delivery, we captured whether the surgeon or anaesthetist was a hospital-employed medical officer or a private GP, the type of anaesthetic given, whether the

procedure was classified as an emergency or elective and whether it was performed during daytime or evening/night-time (16:00–08:00, Monday–Sunday).

Clinical data were collated from registers by research nurses recruited for the purpose of the study. The monthly data were entered into a preset Excel spreadsheet.

### Qualitative data collection

Semistructured interview guides were developed, one for private providers and one for public providers. Qualitative interviews were undertaken by three of the investigators (TD, GS and ED) at the hospitals and district office. The investigators who conducted interviews are all senior researchers with experience in qualitative interviewing. Two were health economists and one a health systems researcher. They had no prior relationships with any of the interviewees. All interviews were conducted in a private consulting room or office and were undertaken in English.

Interviews were undertaken with private GPs who had entered into public service contracts with each of the five hospitals to explore their perceptions of undertaking work in the public sector and the role they played. At each hospital, we also interviewed the hospital managers and public providers (doctors) to explore their perceptions of the public/private interface including clinical decision-making, private GP availability and remuneration models. Types of questions asked to private GPs included: 'What has been your experience of working in government hospitals?' and 'How do you balance your time between private practice and government hospital work?'; types of questions asked to district managers and government doctors included: 'What services are private GPs contracted to provide?', 'How are decisions about caesarean delivery made within the clinical team?' and 'How are private GPs remunerated for their time by the public sector?' During interviews, we also collected information on the total number of hours per month each GP was contracted to the public hospital and used this information to calculate the number of full-time equivalent doctor posts the GPs contributed at each hospital. For example, in hospital C, there were five private GPs each contracted for 25 hours per week. A public sector doctor works 56 hours per week; therefore, in this hospital, the combined private GP hours are equivalent to 2.2 full-time equivalent establishment doctors.

We undertook a total of 23 semistructured qualitative interviews including: 3 district managers (2 male, 1 female), 12 private GPs (7 male, 5 female) and 8 government-employed doctors (4 male, 4 female).

### Quantitative data analysis

Quantitative data from hospital records were analysed in Excel using simple descriptive mean and SD (mean monthly number of deliveries) and proportions (caesarean delivery rate, deliveries undertaken by a private GP, elective caesarean deliveries and caesarean deliveries occurring during daytime or evening/night-time hours)

stratifying by hospital over the 12-month data collation period.

We describe the proportion of total annual caesarean deliveries at each hospital undertaken by private GPs as surgeon or anaesthetist and explore the relationship between type of provider and the caesarean delivery profile (elective vs emergency and timing in terms of daytime hours or during the evening/night) and adverse outcomes (referral to a regional hospital following a caesarean delivery, maternal death, PPH and other procedures performed at caesarean deliveries).

### Qualitative data analysis

Qualitative interviews were digitally recorded and transcribed. A framework analysis[14] approach was applied, which is well suited to implementation research designs. We drew on a framework previously developed by members of our research team which outlines drivers, challenges and required action for obstetric care in preparing for NHI.[15] The interviews were read and reread by members of the team to familiarise ourselves with the content. Emerging categories and major themes were identified and mapped against the framework and quantitative findings. Five members of the research team (TD, ED, GS, SF and YB) read transcripts and met to discuss emerging categories and major themes. The qualitative data enhance contextual understanding of the quantitative outcomes related to caesarean section rates and outcomes by shedding light on provider experiences and perceptions of aspects such as workload and remuneration.

### Patient and public involvement

Patients and/or the public were not involved in the design, or conduct, of this research. We will be disseminating the findings through presentations at each of the participating hospitals, which will be targeted at stakeholder groups including patients and members of the public.

### RESULTS

From the analysis of the qualitative interviews, four main themes were identified, which included: (1) the role played by private GPs contracted to district hospitals; (2) the nature of clinical decision-making regarding caesarean delivery; (3) balancing work between private practice and public sector; and (4) perspectives on remuneration models. These results are reported in narrative form, as quotes, woven within the quantitative results below.

### Hospital characteristics and utilisation of private GPs for caesarean deliveries

The five district hospitals ranged in size from 35 beds in hospital A to 118 beds in hospital E. Over the period 1 April 2021–31 March 2022, the average number of monthly deliveries ranged from 34 in hospital A to 144 in hospital E. None of the hospitals has primary care clinics

in its catchment areas that perform vaginal deliveries other than in the case of emergencies and only during daytime hours. All five hospitals have some involvement of private GPs to assist with the obstetric service but the level of utilisation of their services differed greatly across the hospitals. In hospital A, there are no government-employed doctors and the hospital is run entirely by a private GP practice consisting of five doctors who provide 24-hour cover to the hospital. Across the other hospitals, the number of contracted GPs ranged from one in hospital E to five in hospital C, but this translated into a maximum of 2 full-time equivalent doctors in hospital C and 0.2 in hospital E (table 1).

Online supplemental table 1 shows the profile of deliveries occurring in the hospitals over the 12-month study period. The proportion of normal vaginal deliveries ranged from 74% in hospitals C and E to 85% in hospital D. Assisted vaginal deliveries were uncommon (<3%) across all five hospitals. The caesarean delivery rate ranged from 14% in hospital D to 25% in hospital E. Spinal anaesthesia was used in 95% of all cases.

The use of private GPs as surgeon or anaesthetist for caesarean deliveries differed widely across the five hospitals. Overall, the utilisation of private GPs for anaesthetics was similar to (29% of all caesarean deliveries) the utilisation of private GPs as surgeons (33% of all caesarean deliveries), although in hospital C, more private GPs were used for anaesthetics than for surgery (online supplemental table 1). Interviews with private GPs and hospital-employed doctors spoke to the role that private GPs play in district hospitals (theme 1) and revealed that the skills shortage was perceived to be surgical rather than anaesthetic, and that private GPs make a contribution to training community service doctors in surgical skills for caesarean deliveries as these doctors explained:

> I'm doing more the cutting side. So, I mean, the comserves (community service doctor) can't cut. I am always cutting. There's one comserve on and it's me. So, we cut with the sister. (private GP, hospital B)

> So, in a month I must train all the community service doctors to be safe to do caesarean deliveries and ectopics which is not an easy undertaking. (clinical manager, hospital D)

> That is just the GP that is usually the primary surgeon. I am more of the assistant. I'm still learning to do caesars on my own. So, it's usually just the GPs who are doing the primary surgery. On the anaesthetic side there's another GP that is on call. We're basically an assistant. I'm still learning so I mostly doing primigravida's at the moment. (year 1 medical officer, hospital C)

> I'm on call every second day of my life and every second weekend of my life because we, as the two of us (private GPs), are the only people that can do caesarean deliveries. The rest of the people are either young MOs with little experience or, community

| Table 1 | Characteristics of participating hospitals | | | | |
|---|---|---|---|---|---|
| **Hospital** | **Hospital A** | **Hospital B** | **Hospital C** | **Hospital D** | **Hospital E** |
| Number of beds | 35 | 50 | 75 | 85 | 118 |
| Number of maternity beds (antenatal, labour and postnatal) | 8 | 10 | 12 | 14 | 25 |
| Number of clinics in the hospital catchment that perform vaginal deliveries | 0 | 5 clinics that can do emergency vaginal deliveries if required, no 24-hour service | 0 | 1 community day clinic, 4 clinics and 6 satellite services, none offer 24-hour service, can do emergency vaginal deliveries if required | 10 clinics that can do emergency vaginal deliveries if required, no 24-hour service |
| Mean monthly number of deliveries (SD) | 34 (7.2) | 37 (9.7) | 83 (5.2) | 123 (9.8) | 144 (18.3) |
| Number of full-time hospital-employed doctors | 0 | 6 | 6 | 10 (2 of which are specialist family physicians) | 13 |
| Number of community service doctors included in the total medical staff | 0 | 4 | 2 | 4 | 5 |
| Number of privately contracted GPs to assist with the obstetric service | One practice consisting of 5 GPs providing a 24-hour service undertaking both surgery and anaesthesia | 2 GPs undertaking surgery | 5 (2 for anaesthetics and 3 for surgery) | 4 (2 surgical, 2 anaesthetic); 1 GP has a postgraduate diploma in anaesthetics, 1 GP has a postgraduate diploma in obstetrics, 1 GP is a specialist family physician | 1 GP doing surgery only |
| Number of full-time equivalent private GPs | 0.9 | 0.3 | 2.2 | 0.9 | 0.2 |
| GPs, general practitioners. | | | | | |

service doctors that cannot do that. (private GP, hospital B)

The proportion of caesarean deliveries undertaken by private GPs as the primary surgeon was inversely related to size of hospital and total average monthly deliveries (figure 1). The two smallest hospitals were most reliant on private GPs to perform caesarean deliveries and as size of hospital increased, fewer caesarean deliveries were undertaken by private GPs. The caesarean delivery rate was not higher in hospitals where private GPs carried out a higher proportion of caesarean procedures (figure 1). During interviews, the reasons given for this were that it is generally the hospital-based staff who make the decision about whether a caesarean delivery is required, and the private GP is not involved in that decision. In hospital

A, which is run entirely by private GPs, the midwives managing the labour ward assess progress of labour and call the GPs from their practice if they assess that a caesarean is required. The following quotes illustrate the decision-making around caesarean deliveries (theme 2):

The contract that I have is explicitly for theatre work only. Nothing else. I don't go into labour ward, maternity ward. I get called and I come straight to theatre. The decisions have been made by the medical officers in the ward or in casualty and then I'm the technician. I get called in to do mostly caesars but also ectopic pregnancies and incomplete abortions and that sort of thing. And complications of pregnancy, cervical tears, vaginal tears that sort of thing,

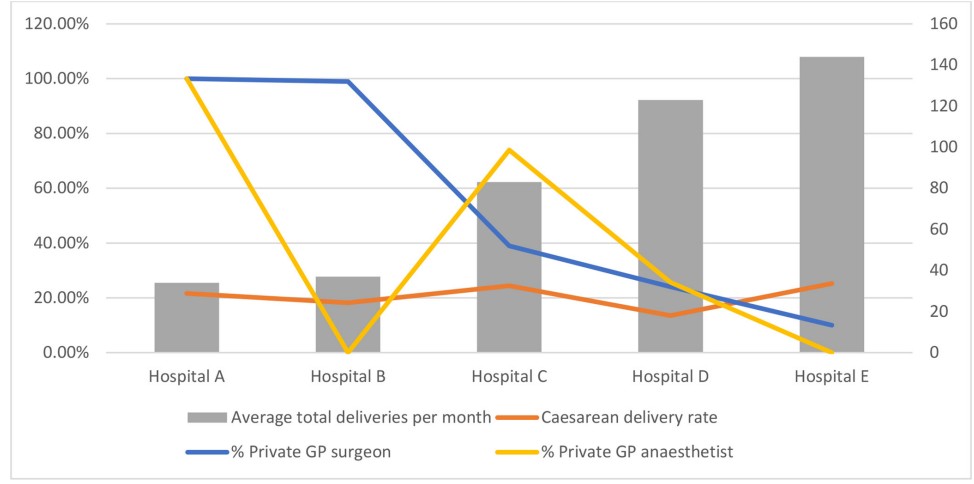

**Figure 1**  Private GP utilisation and caesarean delivery rate. GP, general practitioner.

retained placentas, but it's only theatre work. (private GP, hospital E)

They book them. We don't really see the patient. We do the surgeries for them just in terms of logistics, and then they follow up with the aftercare after that. (private GP, hospital C)

### Profile and outcomes of caesarean deliveries with private GP contracting

The majority (71%) of caesarean deliveries across all five hospitals were emergency procedures (online supplemental table 1). The most common indication for an emergency caesarean delivery was fetal distress or an abnormal Cardiotocography(CTG) (48%) (online supplemental table 2). Among elective caesarean deliveries, the main indication was a previous caesarean delivery (58%). For 5% of emergency caesarean deliveries and 20% of elective caesarean deliveries, the indication was missing from the theatre register. Individual hospital documentation style differed with a higher proportion of elective caesarean deliveries (72%) with a documented indication missing at hospital A, which was serviced only by private GPs, while this was less than 30% at the other four hospitals. However, this did not affect the overall caesarean section rate (online supplemental table 3).

Adverse outcomes following a caesarean delivery indicated by blood loss >1000 mL, maternal death post-caesarean delivery and referral after caesarean delivery to a regional hospital were rare across all hospitals with a 1% prevalence of PPH and 4% of women being transferred to a regional hospital following caesarean delivery. Over the 12-month period of data collection, there were no emergency procedures (hysterectomy, B-lynch suture or re-look laparotomies) performed during or following a caesarean delivery at any of the hospitals (online supplemental table 4).

Figure 2 shows the utilisation of private GPs for caesarean deliveries and the profile and outcomes of these deliveries.

Slightly over half of all caesarean deliveries were performed during public sector working hours (08:00–16:00) but this differed across the hospitals. In hospitals A and B, the majority of caesarean deliveries were performed during evening/night-time (16:00–08:00). In these two hospitals, almost all of the caesarean deliveries are performed by private GPs who consult in their private practices during the day. The challenge of balancing a private GP practice with contractual commitments to the district hospital (theme 3) was explained by the following participants:

So, we come to hospital from eight to ten in the morning. Sometimes it does take longer and you only get down there at half past ten. So, from ten to twelve it's like private practice, then clinics, and then from half past two to half past four, five-ish at the practice again. (private GP, hospital B)

If it was a GP that was covering, they would be a bit reluctant to come in at four to assist us because having their own practice, they might still have practice till five or maybe six sometimes. So, they would maybe push the hour that they would be available to come in, a little bit further on, but, usually, if we call for a caesar, it's usually a foetal distress, we don't just cut to cut if it's not a foetal distress then we would say we could wait another half an hour. (medical officer, hospital E)

There is no noticeable increase in elective caesarean deliveries with a higher proportion of total caesarean deliveries undertaken by private GPs and rates of referral to a regional hospital following caesarean delivery were low irrespective of the utilisation of private GPs.

The funding model used to remunerate private GPs for their work within the public sector is a rate per shift irrespective of the number or type of procedures performed. Several GPs described the remuneration model during interviews (theme 4):

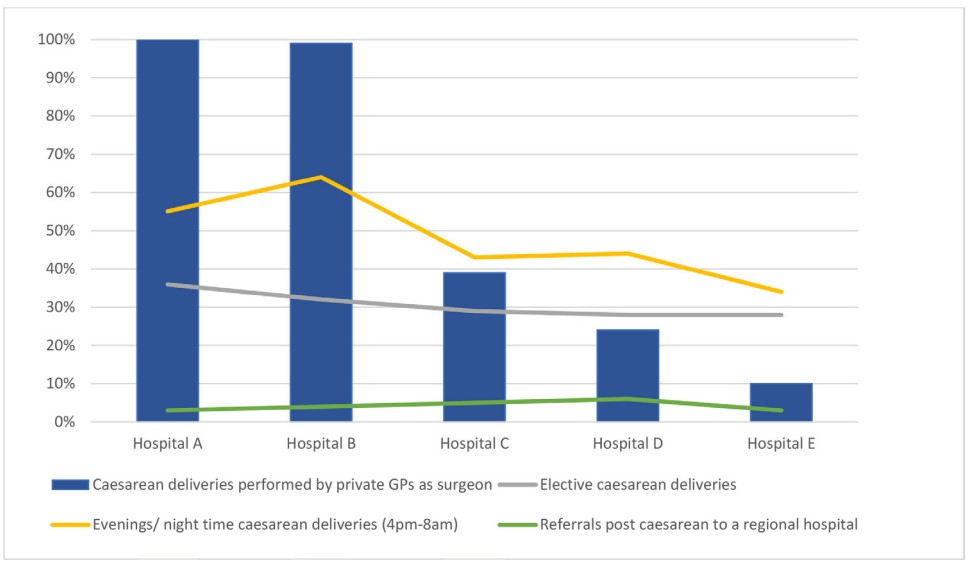

**Figure 2** Utilisation of private GPs and caesarean section profile and outcomes. GPs, general practitioners.

I do one week per month from Monday night, four p.m. till eight a.m. the next Monday and the remuneration is not affected by the number of call outs. It's a flat rate. Whether I get called once in that week or twenty times in that week, the remuneration is the same. Payment every month exactly the same. If I'm lucky, I get called out only once or twice for the week, but, a week ago, I was called out three times in one night. An ectopic pregnancy, abruptio and, obstructed labour. Just like that. It's four hours in theatre. (private GP, hospital E)

You never know how your hours are going to turn out for that week. So, say this week I have a very good week, I only have to be there from eight till eleven. They never phoned me during the night; they never phoned me during my day, the guy who's on next week they're phoning the whole time because it's a caesarean after caesarean. Every night there is an ectopic or a third-degree tear or something so you can't really, you can't really compare the two calls, but everyone still gets the same fixed rate. (private GP, hospital C)

Because the sessional guys (private GPs) are just for the sessions. So, I mean they don't get paid extra if they do ten or one (caesarean deliveries). (clinical manager, hospital D)

## DISCUSSION

This health systems research study explored the utilisation of private GPs to provide caesarean delivery services in rural district hospitals in the Western Cape. We found that the extent of utilisation of private GPs was inversely related to the size of hospital and number of deliveries. The two larger hospitals with over 100 monthly deliveries had a critical mass of doctors to cover the roster and used private GPs less frequently.

In terms of the delivery profile, we found an overall caesarean delivery rate of 21%. This is higher than the WHO-recommended rate of 5%–15% of all births[16] but lower than the public sector rate for the Western Cape of 29%.[2] Caesarean delivery rates in district hospitals are lower than the national rate because such hospitals refer women with complications such as two previous caesarean deliveries, pre-eclampsia, placenta praevia, etc to regional or tertiary hospitals. Also, they deliver low-risk women having normal vaginal deliveries because they do not have surrounding 24-hour midwife units, so these low-risk births are included in the denominator. Hospital D's caesarean delivery rate (14%) may be lower because the service was closed for several years following an extensive fire in March 2017, with all caesarean deliveries being done elsewhere. The caesarean delivery service resumed again in February 2021 just prior to the start of data collection for this study. This hospital is also unique in being the only one of the five with specialist family physicians overseeing the clinical services. All caesarean

deliveries are discussed 24/7 with a family physician prior to being performed, unless such a discussion may delay safe delivery, which may also contribute to the lower rate of caesarean deliveries.

We did not find an increased caesarean delivery rate in hospitals where a high proportion of caesarean deliveries were performed by private GPs compared with government-employed generalist doctors and no increase in caesarean deliveries performed for elective indications. The public–private partnership arrangements in these hospitals are such that the decision-making regarding the need for a caesarean delivery is undertaken by a hospital employee (doctor or midwife) and the private GP is notified once the decision has been made. However, we did find a higher proportion of elective caesarean deliveries without stated indication and a higher proportion performed after 16:00 (during evenings and at night) at hospitals more reliant on private GPs. Balancing time between private practice and public sector work may pose a risk to public hospitals where there are delays in private GP availability in emergency situations.

The remuneration model for GP contracting, with a fixed rate irrespective of number of caesarean deliveries should disincentivise overservicing. A public–private partnership programme in Gujarat state, India provides private obstetricians with a fixed sum per 100 births to provide care to eligible women below the poverty line. The remuneration model was specifically chosen as a disincentive to prevent unnecessary caesarean deliveries.[17] By comparison in Chile, where women covered under public health insurance can choose to give birth within a selected group of private health facilities, the caesarean delivery rate was found to be 71% compared with 26% among publicly insured women giving birth in public health facilities. Publicly insured women choosing to give birth in a private facility receive a voucher that covers the costs of the delivery except for a 25% co-payment, the sum of which is dependent on the mode of delivery, which is the responsibility of the patient.[18]

In terms of caesarean delivery outcomes, our results suggest that in rural district hospitals using experienced private GPs to perform caesarean deliveries, there was no trend towards increased adverse outcomes. The rate of complications post-caesarean section was low across all five hospitals, which is an indication of safe caesarean section deliveries overall in this setting.

There was one maternal death following caesarean delivery in the 1-year period of review due to eclampsia, a 1% rate of PPH at caesarean delivery and a very low rate of referral to a regional hospital following caesarean delivery. The rate of PPH at caesarean delivery was low. This is likely to reflect the lower risk caesarean deliveries that were performed at district hospitals, good labour ward management with early recourse to caesarean delivery and surgical skills. It is possible that some women were referred to higher-level hospitals for bleeding after caesarean delivery, but this information is not available because individual folders were not reviewed, and the

numbers would be few because overall referral numbers were low.

In SA, district hospitals account for 38% of public sector caesarean deliveries and the CFR for caesarean deliveries at this level of care is 62 per 100 000 caesarean deliveries.[2] Since these hospitals do not have specialist obstetricians or anaesthetists, surgical and anaesthetic skills training and supervision are critical. Deaths due to bleeding at or following a caesarean delivery have been increasing in SA and lack of skilled doctors has been identified as an important avoidable factor.[19] This model of a public–private partnership makes an important contribution to skills development in rural district hospitals through drawing on the skills and experience of private GPs, and in some hospitals private family physicians, to perform caesarean deliveries alongside hospital-based medical officers, many of whom are undertaking community service or recently qualified.

Balancing access with safety is a critical factor in obstetric care. Ideally, facilities that can provide care for uncomplicated births should be close to women's homes and within reasonable distance of a district hospital that can perform caesarean deliveries. For small district hospitals with a low monthly delivery rate, maintaining skills to deal with emergencies is a challenge. To be safe and cost-effective, it has been recommended that district hospitals would need to perform between 500 and 1200 deliveries per year.[20] There should also always be a minimum of two doctors on call (one for anaesthesia and one for surgery), so that caesarean deliveries may be performed on a 24-hour-a-day, 7-day-a-week basis.[20] Hospitals A and B in this study performed less than 500 deliveries per year and had the highest utilisation of private GPs. It would be more costly for these hospitals to employ permanent doctors on the establishment and this model of public–private contracting makes financial sense and provides access for women since the nearest regional hospital is roughly 150 km from the district.

Considering broader implications of these findings for universal healthcare, it is clear that experienced private GPs can play an important role in contributing much needed surgical and anaesthetic skills for safe caesarean deliveries in rural district hospitals. This is a model that could be considered for other provinces in SA and other LMICs provided clinical decisions are driven by accepted public sector protocols and available support from specialists at referral regional hospitals can be ensured. Public–private partnerships have been criticised for being a more urban context option, but this is often on the assumption that the private providers will be obstetric specialists and this is the model that has been applied in settings such as India.[9 10 21] By contrast, in African countries such as Mozambique and Tanzania, clinical officers (healthcare providers trained to perform tasks usually undertaken by doctors) have been trained to perform caesarean deliveries. A meta-analysis reported no significant differences in outcomes of caesarean deliveries performed by clinical officers compared with medical doctors.[22] Different approaches to enable access to safe caesarean delivery are therefore needed for different contexts and experienced private GPs are a resource for rural district hospitals to consider. The sustainability of this model depends on the surgical and anaesthetic skills of future generations of GPs. The GPs skilled in obstetric surgery and anaesthesia in rural districts are mostly of an older age, many of whom were previously district surgeons (a historical model of contracting private GPs within rural districts to provide curative primary care services within public sector health facilities).[23] More recently qualified GPs may have similar skills to their public sector medical officer equivalents. The scale-up of this model therefore has implications for undergraduate basic surgical and anaesthetics skills training programmes.

## LIMITATIONS

We collected obstetric data from registers and did not perform individual patient folder reviews. The proportion of caesarean delivery complications could have been higher if individual folders were reviewed. Post-caesarean sepsis, thromboembolism or postoperative bleeding that was detected in the postnatal ward, for example, could not be identified with the study methodology. Where no indication was listed in the theatre register for a caesarean delivery, this information would have been found had individual folders been reviewed. Furthermore, although a low proportion of women were referred following caesarean delivery, this group could have experienced morbidity or mortality but were not followed up to determine the reasons for referral or their outcomes at the regional referral hospital. We also did not collect data on neonatal outcomes, which is an important indicator of the quality and safety of caesarean delivery.

## CONCLUSION

The findings of this study suggest that private GPs can play an important role in filling gaps and expanding quality care in rural public facilities that have insufficient obstetric skills and expertise. In addition, provided that the broader systems and institutional framework to ensure compliance with clinical guidelines and protocols are in place, this can be done in a safe and effective manner, and could be seen as a prototype for NHI in rural contexts.

**Author affiliations**
[1]Health Systems Research Unit, South African Medical Research Council, Tygerberg, South Africa
[2]Paediatrics and Child Health, University of Cape Town, Cape Town, South Africa
[3]School of Public Health, University of the Western Cape, Cape Town, South Africa
[4]Health Economics Unit, University of Cape Town, Cape Town, South Africa
[5]NMG Consultants and Actuaries, Cape Town, South Africa
[6]Western Cape Government: Health and Wellness, Cape Town, South Africa
[7]Department of Obstetrics and Gynaecology, University of Cape Town, Observatory, Western Cape, South Africa

**Acknowledgements** We thank all the health professionals who took time to participate in our interviews and to the hospital managers for facilitating access to the obstetric registers. We also thank Vishal Brijlal for his comments on a draft of this manuscript.

**Contributors** TD, GS, ED and SF conceptualised the study. TD, GS, ED and YB analysed the data. TD, GS, ED, AH, SF and YB contributed to the interpretation of the data, writing the paper and reviewing the content. TD is the guarantor.

**Funding** The study was funded by the Bill and Melinda Gates Foundation (grant agreement ID INV-023276).

**Competing interests** None declared.

**Patient and public involvement** Patients and/or the public were not involved in the design, or conduct, or reporting, or dissemination plans of this research.

**Patient consent for publication** Not required.

**Ethics approval** Ethics approval was granted by the South African Medical Research Council Human Research Ethics Committee (approval number EC003-2/2021) and permission was also granted by the Western Cape Provincial Department of Health and Wellness (approval number WC_202103_017). No personal-identifying information was captured from hospital records, only aggregated summary information. All health professionals who were approached to participate in an interview were read an information sheet outlining the expectations in terms of length of the discussion, the voluntary nature of the participation and measures to ensure confidentiality. Health professionals who agreed to participate signed an informed consent form.

**Provenance and peer review** Not commissioned; externally peer reviewed.

**Data availability statement** Data are available upon reasonable request. The aggregated routine data are available on request to the corresponding author.

**ORCID iDs**
Tanya Doherty http://orcid.org/0000-0003-1592-0080
Geetesh Solanki http://orcid.org/0000-0002-8512-3119

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
