## [Reviewer comments · BMJ Open]

ARTICLE DETAILS

TITLE (PROVISIONAL)	Utilisation of private general practitioners to provide caesarean deliveries in five rural district public hospitals in South Africa: a mixed-method study
AUTHORS	Doherty, Tanya; Solanki, Geetesh; Daviaud, Emmanuelle; Bartmann, Yvandi; Hawkrigde, Anthony; Fawcus, Sue

VERSION 1 – REVIEW

REVIEWER	Bijlmakers, Leon Radboud University Medical Center, Radboud Institute for Health Sciences (RIHS), Health Evidence
REVIEW RETURNED	15-Sep-2022

GENERAL COMMENTS	I am glad for the opportunity to review this manuscript on a very relevant study topic, with an appropriate study design. It reads well. I have several observations and suggestions which you may want to consider to improve the paper. Objective, as stated in the Abstract, “Researching how different models of maternity care in private and public facilities can work together to reduce We aimed to document ...” : 1. It’s a bit weird to talk of models that would need to work together. Please rephrase.2. From my understanding your paper is not about private and public health facilities that (can) work together, but rather public facilities contracting private GPs as primary surgeons or anaesthetists – which is much more specific. The positioning of and possible implications for private facilities to which these private GPs may be affiliated (e.g. absence from work) is not the object of this study.3. In the paper itself the objective is phrased somewhat differently: I suggest you stick to one and the same phrase. Conclusion, as formulated in the Abstract: “The findings of this study suggest that private GPs can play an important role in filling “gaps” and expanding quality care in the public sector in areas where the total number of doctors or skills and expertise are in short supply, such as in rural areas.” I suggest to rephrase as follows: ... in filling gaps and expanding quality care in (rural) public facilities that have insufficient obstetric skills and expertise (or manpower). Two new findings are highlighted on page 4. You would need to be a bit more modest, given the nature and small scale of this study. Such statements would need to be validated with a stronger study design that allows a comparison of the obstetric performance of
--

	public health facilities with and without contracted private GPs, or before and after contracting them. Introduction  • First paragraph: it would be good to also present C/S rates, CFR and haemorrhage rates in public facilities compared to private facilities? Are there any such data? • “Improved clinical governance and skills development in the public sector are critical to reducing this disparity, but these efforts would need to be complemented by the mobilization of skills from the private sector given that the bulk of the resources lie in the private sector.4 “ This sentence puzzles me a bit: (a) are you saying that the private sector (or private facilities) performs better than the public sector, or are you simply referring to the (presumed) limited volume of resources that are available in the public sector? ; and (b) do you mean resources in general, or human resources (skilled manpower)? Or obstetric specialists in particular? Methods section  • I like the last paragraph of the study setting (pages 7, lines 21-38) which nicely describe the existing modalities for hiring of private GPs by public health facilities. What I am missing though is: (a) how are these private GPs remunerated? (i.e. do they get paid per case, or per day/hours worked?); and (b) is there any performance-related or output-related financial reward for the workers employed by the public facilities (e.g. per delivery performed). I guess the latter is not the case. • Then upon reading further I found in the Results section (page 15) : “The funding models used to remunerate private GPs for their work within the public sector is a rate per shift irrespective of the number or type of procedures performed and therefore there is no undue incentive to perform unnecessary elective caesarean deliveries.” The first part of this sentence comes a bit as a surprise: I suggest you modify the study aim a little. Instead of saying “We aimed to document the utilisation of private general practitioners ...” you could say: “... to document the frequency and the way in which public facilities utilise private GPs ...”. The latter part of the above sentence (starting with ‘therefore’) is definitely not something you should say here, but you can bring it up in the Discussion section. It is good to be aware of possible undesired side-effects of such contracting, but from your study results you cannot conclude with certainty that there are no undue incentives to perform CS/surgery. Results section  • Page 12, “Overall, the utilisation of private GPs for anaesthetics was less (29% of all caesarean deliveries) than the utilisation of private GPs as surgeons (33% of all caesarean deliveries) ...” : I would rather say that these utilisation rates did not differ much. You could do a statistical test to see if the observed difference is statistically significant. • “Interviews with both private and hospital employed doctors revealed that the skills shortage is mainly with surgical rather than anaesthetic skills, ...”. I suggest to qualify this: the skills shortage IS PERCEIVED as mainly of a surgical nature rather than for anaesthesia. • In relation to the statement “... that private GPs make a large contribution to training community service doctors in caesarean
--	---

	deliveries ...”: is it that private GPs do not (or rarely) train local staff to apply anaesthesia? Or are they not allowed to do that? Or is nobody interested in acquiring those skills? Or didn’t the researchers further explore this? Discussion  • The statement that “... private GPs can safely perform caesarean deliveries in public district hospitals without compromising safety” is too strong. I object to the implicit suggestion that one maternal death following caesarean delivery in the one-year period of review is little. The number of C-sections in the five hospitals included in the study is not that high; and with a “case fatality rate for caesarean deliveries at this level of care (of) 62 per 100,000 caesarean deliveries” you wouldn’t expect more than one maternal death. Conclusion  • Most of this section is actually a discussion. And you seem to go much further than what can be concluded from this study. While I agree that there is scope for further studies I would not advise you venture into other types/modalities of public private collaboration. While it is probably true that there is a need to review (other) service delivery models for obstetric care in the private sector prior to extending public-private contracting to specialist obstetricians, this cannot be concluded from the present study. (The fact that caesarean delivery rates amongst private specialist obstetricians in South Africa are high (6) , far higher than global averages or recommended norms, is worrisome indeed.) Textual  • Please be a little more consistent in the use of the terms of (health) services and (health) facilities. For example, on page 8, line 21, “The public health service is permitted to contract the services of private providers where needed.” - please change to public health facilities. • Top of page 14, “The proportion of caesarean deliveries undertaken by private GPs as the primary surgeon showed a decreasing trend as size of hospital and total average monthly deliveries increased ...”: ‘trend’ is not the right term to use, as it normally refers to a change over time. I would suggest: the proportion ... was inversely related to ... References: for the first one you need to mention WHO at the beginning.
--	---

REVIEWER	Hoxha, Ilir Kolegjin Heimerer
REVIEW RETURNED	26-Oct-2022

GENERAL COMMENTS	Comments This paper examines the utilisation of private general practitioners to provide caesarean deliveries and other maternal outcomes in five rural district public hospitals. They were interested to learn how public-private contracting is working in light of current healthcare reform by focusing on obstetric services. But I find this study descriptive as the authors intended. It provides little material to assess the health care reform impact. It’s not well designed, executed or presented. Specific details below. Title
---

	It could be improved and shorter. Suggest a more compact title. Introduction The intro seems very interesting and relevant but seems very general to me. While it's useful to know all that information about healthcare reform. I would expect some serious introduction as to why studying contracting, primary care providers and obstetric outcomes is of relevance for that. There is little about this. As a reader, I find it hard to connect the introduction with the main research hypothesis. Methods Overall could have been clearer and better structured. It contains a lot of details and useful info. But it's not that well-written and clear. May be useful to be clearer on specific methods (quantitative and qualitative) of data collection and analysis. Maybe there can be some common subheadings for both... but other parts have to be clear for each method used. On quantitative data collection. Minor point but seems relevant to me. Was clinical data collected from participants? Was it collected or was it processed/monitored? I need clarification on this. If data was taken from existing registers, for me is just data processing... its not data collection... from users... so this has to be clear... even if you are using data collection as a term... which would be fine... There is a glimpse of the description of outcomes of interest, like the CS rate... but it's not clear me what were outcomes of interest in using the quantitative part... what are the themes of interest in qualitative research... no description of details/content of the semi-structure instrument. What things it was designed to measure? Results After reading the results, I find that this study is a descriptive study, as the authors explain in the methods section. But it only provides a little help in understanding reform or the impact of reform on the outcomes of interest.
--	---

REVIEWER	Adam, Yasmin Wits University
REVIEW RETURNED	31-Oct-2022

GENERAL COMMENTS	Thank you for asking me to review the article entitled "Utilisation of private general practitioners to provide caesarean deliveries, and maternal outcomes in five rural district public hospitals: Insights for public-private contracting" This is an interesting and important subject in the SA context. The paper is well written and easy to read and intends to provide information toward health systems research. The study is set in the Western Cape where private GP's are employed in order to provide obstetric services. The title is misleading as I am not sure that maternal outcomes were explored in this study. There were no hysterectomies or B-lynches, but does not discuss blood transfusion and the reasons for transfer to regional hospitals. They did not collect individual patient data- so one cannot be sure that there were no complications in the 1st 24 hours and more. The main aim is to document the use of Private GP's in the public sector and this is what the study does. It does not explore the advantages of this system from full time staff although it states that midwives and MO were interviewed. The Introduction is well written and explains the rationale for doing the study. The methods
---

	is clearly outlined. The data analysis needs some clarification. The results are well set out, but do not address all the objectives that the paper set out to answer see my comments on attached file
--	---

REVIEWER	Gebhardt, Stefan Stellenbosch University
REVIEW RETURNED	04-Nov-2022

GENERAL COMMENTS	This is a well written paper and important for any country exploring models for private-public initiatives. The major limitation is acknowledged, and that is that there was no folder review, so the outcomes could very well be different if actual reasons for referral or re-admission were scrutinized. There was one maternal death amongst the 5048 deliveries- was this death related to CS and/or difficulty during CS, or unrelated to any surgery or anesthetics? Another limitation worth mentioning is that GPs skilled in obstetric practice in rural SA are mostly from an older generation, and the younger/newer doctors have similar lack of skills than their first year public sector medical officer counterparts, as there is no specific GP obstetrics training required. Will this model also work in a setting where the newer generation of GPs are not skilled in safe CS or safe anesthesia? In smaller SA towns, where GPs also do private obstetrics, these private patients usually deliver in the public sector hospital as it is the only hospital in town. When this 'private' patient develops complications and can no longer afford private care, she is handed over to the public sector (but may still receive a 'free' CS from her GP if he/she is on call that night). This is a potential loophole that can be exploited, as the GP gets paid for private consultations and for the hospital work. Was this option discussed in the interviews?
--

VERSION 1 – AUTHOR RESPONSE

Reviewer: 1

Dr. Leon Bijlmakers, Radboud University Medical Center, Radboud Institute for Health Sciences (RIHS)

Comments to the Author:

I am glad for the opportunity to review this manuscript on a very relevant study topic, with an appropriate study design. It reads well. I have several observations and suggestions which you may want to consider to improve the paper.

Objective, as stated in the Abstract, “Researching how different models of maternity care in private and public facilities can work together to reduce We aimed to document ...” :

1. It's a bit weird to talk of models that would need to work together. Please rephrase.
2. From my understanding your paper is not about private and public health facilities that (can) work together, but rather public facilities contracting private GPs as primary surgeons or anaesthetists – which is much more specific. The positioning of and possible implications for private facilities to which these private GPs may be affiliated (e.g. absence from work) is not the object of this study.
3. In the paper itself the objective is phrased somewhat differently: I suggest you stick to one and the same phrase.

Response: Thank you we agree and have revised the objective in the abstract and the paper taking into the account the comments from all four reviewers (page 2 and 5)

Conclusion, as formulated in the Abstract: “The findings of this study suggest that private GPs can play an important role in filling “gaps” and expanding quality care in the public sector in areas where the total number of doctors or skills and expertise are in short supply, such as in rural areas.” I suggest to rephrase as follows: ... in filling gaps and expanding quality care in (rural) public facilities that have insufficient obstetric skills and expertise (or manpower).

Response: We agree and have made the suggested change (page 2)

Two new findings are highlighted on page 4. You would need to be a bit more modest, given the nature and small scale of this study. Such statements would need to be validated with a stronger study design that allows a comparison of the obstetric performance of public health facilities with and without contracted private GPs, or before and after contracting them.

Response: Thank you we have removed this statement.

Introduction

- First paragraph: it would be good to also present C/S rates, CFR and haemorrhage rates in public facilities compared to private facilities? Are there any such data?

Response: We have added the suggested indicators for both sectors where this is available (page 4).

- “Improved clinical governance and skills development in the public sector are critical to reducing this disparity, but these efforts would need to be complemented by the mobilization of skills from the private sector given that the bulk of the resources lie in the private sector.⁴” This sentence puzzles me a bit: (a) are you saying that the private sector (or private facilities) performs better than the public sector, or are you simply referring to the (presumed) limited volume of resources that are available in the public sector? ; and (b) do you mean resources in general, or human resources (skilled manpower)? Or obstetric specialists in particular?

Response: We have amended this sentence to make it clearer (page 4)

Methods section

- I like the last paragraph of the study setting (pages 7, lines 21-38) which nicely describe the existing modalities for hiring of private GPs by public health facilities. What I am missing though is: (a) how are these private GPs remunerated? (i.e. do they get paid per case, or per day/hours worked?); and (b) is there any performance-related or output-related financial reward for the workers employed by the public facilities (e.g. per delivery performed). I guess the latter is not the case.

Response: We have added to study setting a description of the remuneration model (page 7)

- Then upon reading further I found in the Results section (page 15) : “The funding models used to remunerate private GPs for their work within the public sector is a rate per shift irrespective of the number or type of procedures performed and therefore there is no undue incentive to perform unnecessary elective caesarean deliveries.” The first part of this sentence comes a bit as a surprise: I suggest you modify the study aim a little. Instead of saying “We aimed to document the utilisation of private general practitioners ...” you could say: “... to document the frequency and the way in which public facilities utilise private GPs ...”.

Response: Thank you we agree and have revised the study aim in the abstract and the paper taking into the account the comments from all four reviewers (page 2 and 5)

The latter part of the above sentence (starting with ‘therefore’) is definitely not something you should say here, but you can bring it up in the Discussion section. It is good to be aware of possible undesired side-effects of such contracting, but from your study results you cannot conclude with certainty that there are no undue incentives to perform CS/surgery.

Response: We agree and have removed this statement.

Results section

- Page 12, “Overall, the utilisation of private GPs for anaesthetics was less (29% of all caesarean deliveries) than the utilisation of private GPs as surgeons (33% of all caesarean deliveries) ...” : I would rather say that these utilisation rates did not differ much. You could do a statistical test to see if the observed difference is statistically significant.

Response: We agree and have revised this text (page 12)

- “Interviews with both private and hospital employed doctors revealed that the skills shortage is mainly with surgical rather than anaesthetic skills, ...”. I suggest to qualify this: the skills shortage IS PERCEIVED as mainly of a surgical nature rather than for anaesthesia.

Response: We agree and have revised this text (page 12).

- In relation to the statement “... that private GPs make a large contribution to training community service doctors in caesarean deliveries ...”: is it that private GPs do not (or rarely) train local staff to apply anaesthesia? Or are they not allowed to do that? Or is nobody interested in acquiring those skills? Or didn't the researchers further explore this?

Response: We have clarified that the on-site training was related to surgical skills (page 12).

Discussion

- The statement that “... private GPs can safely perform caesarean deliveries in public district hospitals without compromising safety” is too strong. I object to the implicit suggestion that one maternal death following caesarean delivery in the one-year period of review is little. The number of C-sections in the five hospitals included in the study is not that high; and with a “case fatality rate for caesarean deliveries at this level of care (of) 62 per 100,000 caesarean deliveries” you wouldn't expect more than one maternal death.

Response: We agree and have revised this sentence (page 18).

Conclusion

- Most of this section is actually a discussion. And you seem to go much further than what can be concluded from this study. While I agree that there is scope for further studies I would not advise you venture into other types/modalities of public private collaboration. While it is probably true that there is a need to review (other) service delivery models for obstetric care in the private sector prior to extending public-private contracting to specialist obstetricians, this cannot be concluded from the present study. (The fact that caesarean delivery rates amongst private specialist obstetricians in South Africa are high (6) , far higher than global averages or recommended norms, is worrisome indeed.)

Response: Thank you we agree and have removed this statement and revised the conclusion (page 21).

Textual

- Please be a little more consistent in the use of the terms of (health) services and (health) facilities. For example, on page 8, line 21, “The public health service is permitted to contract the services of private providers where needed.” - please change to public health facilities.

Response: We have made the suggested change (page 7).

- Top of page 14, “The proportion of caesarean deliveries undertaken by private GPs as the primary surgeon showed a decreasing trend as size of hospital and total average monthly deliveries increased ...”: ‘trend’ is not the right term to use, as it normally refers to a change over time. I would suggest: the proportion ... was inversely related to ...

Response: We agree and have made the suggested change (page 14).

References: for the first one you need to mention WHO at the beginning.

Response: Thank you we have corrected this (page 23).

Reviewer: 2

Dr. Ilir Hoxha, Kolegjin Heimerer, Dartmouth College Geisel School of Medicine

Comments to the Author:

Comments

This paper examines the utilisation of private general practitioners to provide caesarean deliveries and other maternal outcomes in five rural district public hospitals. They were interested to learn how public-private contracting is working in light of current healthcare reform by focusing on obstetric services. But I find this study descriptive as the authors intended. It provides little material to assess the health care reform impact. It's not well designed, executed or presented. Specific details below.

Response: We have clarified in the introduction that the National Health Insurance has not yet been implemented it is still in the policy development stage (page 4). We carried out a broader research project to examine various aspects of the contracting arrangements related to the use of the private GP's to provide caesarean deliveries in rural public hospitals with a view to informing the development of public private arrangements for improving obstetric and maternal outcomes in rural areas in South Africa and LMICs more broadly. The aim of this paper was to document our findings on the utilisation of the GP's, the profile and outcomes of the deliveries carried out by them and insights the findings provide for broader public private contracting for obstetric care. We have revised the study objective to clarify the scope of the research (page 2 and 5)

Title

It could be improved and shorter. Suggest a more compact title.

Response: We have shortened and revised the title (page 1).

Introduction

The intro seems very interesting and relevant but seems very general to me. While it's useful to know all that information about healthcare reform. I would expect some serious introduction as to why studying contracting, primary care providers and obstetric outcomes is of relevance for that. There is little about this. As a reader, I find it hard to connect the introduction with the main research hypothesis.

Response: We have revised the introduction to make the argument why research on private sector contracting in the field of obstetric care is important (page 4 and 5).

Methods

Overall could have been clearer and better structured. It contains a lot of details and useful info. But it's not that well-written and clear. May be useful to be clearer on specific methods (quantitative and qualitative) of data collection and analysis. Maybe there can be some common subheadings for both... but other parts have to be clear for each method used.

Response: We have created separate subheadings for qualitative and quantitative data collection and analysis to make these methods clearer (page 7-9).

On quantitative data collection. Minor point but seems relevant to me. Was clinical data collected from participants? Was it collected or was it processed/monitored? I need clarification on this. If data was taken from existing registers, for me is just data processing... its not data collection... from users... so this has to be clear... even if you are using data collection as a term... which would be fine...

Response: We have clarified that the clinical data was collated from existing registers (page 8).

There is a glimpse of the description of outcomes of interest, like the CS rate... but it's not clear me what were outcomes of interest in using the quantitative part...

Response: we have listed all the outcomes that could be collected from review of theatre and delivery registers, given that individual patient files were not examined (page 7).

what are the themes of interest in qualitative research... no description of details/content of the semi-structure instrument. What things it was designed to measure?

Response: We have included examples of questions included in the interview guides (page 8) and an overview of the main themes from the qualitative analysis (page 9).

Results

After reading the results, I find that this study is a descriptive study, as the authors explain in the methods section. But it only provides a little help in understanding reform or the impact of reform on the outcomes of interest.

Response: We appreciate this comment. As clarified above and through revisions in the paper, the National Health Insurance has not yet been implemented and is still in the policy development stage. The aim of this research was to inform the development of contracting arrangements for the proposed reforms rather than to assess the impact of the reforms.

Reviewer: 3

Yasmin Adam, Wits University

Comments to the Author:

Thank you for asking me to review the article entitled "Utilisation of private general practitioners to provide caesarean deliveries, and maternal outcomes in five rural district public hospitals: Insights for public-private contracting"

This is an interesting and important subject in the SA context. The paper is well written and easy to read and intends to provide information toward health systems research. The study is set in the Western Cape where private GP's are employed in order to provide obstetric services. The title is misleading as I am not sure that maternal outcomes were explored in this study. There were no hysterectomies or B-lynches, but does not discuss blood transfusion and the reasons for transfer to regional hospitals. They did not collect individual patient data- so one cannot be sure that there were no complications in the 1st 24 hours and more.

The main aim is to document the use of Private GP's in the public sector and this is what the study does. It does not explore the advantages of this system from full time staff although it states that midwives and MO were interviewed. The Introduction is well written and explains the rationale for doing the study. The methods is clearly outlined. The data analysis needs some clarification. The results are well set out, but do not address all the objectives that the paper set out to answer see my comments on attached file

Comments:

Title- The title is misleading as I am not sure that maternal outcomes were explored in this study.

Response: We agree with this comment and have removed 'maternal outcomes' from the title (page 1).

There was no collection of individual data and no data on the hospital stay. Sepsis is a huge concern and this can only be addressed with individually collected data- this should be explained in the limitations in the discussion.

Response: We have listed the limitation of not collecting individual patient data in the limitations section (page 21). The proportion of caesarean delivery complications could have been higher if individual folders were reviewed. Post caesarean sepsis, thromboembolism, or postoperative bleeding that was detected in the postnatal ward for example could not be identified with the study methodology. Furthermore, although a low proportion of women were referred following caesarean delivery, this group could have experienced morbidity or mortality but were not followed up to determine the reasons for referral or their outcomes at the regional referral hospital.

Abstract- please see comments on the rest of the paper and change the abstract accordingly
Conclusion in the abstract- you did not look at quality of care- so it is incorrect to say that Private GP's can provide good quality of care. It is also incorrect to say Private GP's are an underappreciated resource
as your study did not look at this.

Response: We agree and have amended these conclusions (page 2)

What are the new findings:

Page 4 – line 26. It is incorrect that say that private GP's can perform cs safely? It is only this set of GP's who may have been trained differently from the Comm servs. They may have done O&G for several years or anaesthesia for several years prior to becoming GP's. This statement looks like any GP's may be employed in obstetrics. Safety of caesarean section should also consider the neonatal Outcomes

Response: We agree and have removed the statement regarding safety. We have also added to the limitations that we did not collect neonatal outcomes (page 21)

Page 4- Line 32. What are the new findings-

I think this should read that the study documented that Private GP's are a resource- not that it is under appreciated.

Response: We agree and have made this change (page 20).

Methods

Why is Sunday not "after hours"

Response: Our interest was in day time versus night time since private GPs are often used more at night. We have changed the terminology to reflect evening/night time rather than 'after hours' (page 7)

Study setting:

It would be more important to also say what the referral criteria is to refer to a District Hospital. Not just an explanation of staffing. Which patient are referred to a district hospital and which patients have to be referred to a regional hospital- so that a reader from another system can understand

Response: We have added further details on the health system as it pertains to obstetric care and criteria for referral (page 6 &7).

Qualitative data collection:

Page 8- line 24. The interviews were undertaken to explore the following: to explore their

perceptions of the public/ private interface including contracting procedures, team work and clinical governance. – I don't see this in the results?

Response: We have revised this description to better align with the qualitative findings (page 8).

Data analysis

Descriptive statistics using means ? and frequencies. This is unclear.

Response: We have revised this description to provide greater clarity (page 9).

Results:

Page 9. Line 45. The different hospitals are referred to as A, B, etc. in the table (supplementary) they are named. Please be consistent.

Response: Thank you this was an error which we have corrected.

Table 1- What is average number of deliveries? If this is a mean then a SD for each is necessary.

Response: Yes this is the mean monthly deliveries. We have corrected this in the table and added the standard deviation (table 1).

Table 1- are there interns at these hospitals?

Response: No there are no intern doctors at these district hospitals. We have clarified this on page 6.

Table 1- If you are stating the number doing obstetrics and anaesthetics for some hospitals- then you should state it for all.

Response: we have added these details for all hospitals to table 1.

Does surgery include general surgery? Or only O&G related surgery?

Response: Most of the private GP time is spent on surgery for caesarean delivery but they may also be called for an ectopic pregnancy or other obstetric emergency surgery. This has been described on page 7.

The last row in the table is unclear- either explain how this was calculated in the methods or below the table.

Response: we have clarified the calculation of fulltime equivalent doctors on page 8.

Table 2/ page 12 Line 3-4- unassisted vaginal deliveries is not the correct terminology. It may also be confused with "unattended". An assisted delivery is proper terminology for a vacuum or forceps otherwise NVD should be used. Also address in the table.

Response: We have corrected the terminology to NVD in the text (page 12 and table 2).

Page 12 – Line 18/19 – the exact words from the interviews say nothing about anaesthetic skills It's a huge problem if these doctors are just technicians. It is more important to decide when and who needs a cs then just doing it. What about the hospital which is run by GP's. who makes the decision here.

Response: We have clarified the decision-making process around caesarean delivery in hospital A (page 14).

Page 15- line 27- this is important when private GP's cover emergency work. If a cs is booked and it is an emergency it should not wait. This should be discussed. Whilst the use of private GP's has provided services- shared time may be a potential problem.

Response: We have added to the discussion the potential risk to public sector hospitals of delays in GP availability in emergency situations (page 18).

Page 14-15 last paragraph of page 14. Low complication rate in all of the district hospitals even where the full time doctors – so an indication of safe cs rate overall at this level- not a reflection on GP surgeons vs. full time doctors (who are not Specialist Obstetricians)

Response: We have added this comment that we found a safe caesarean delivery profile overall across these hospitals (page 19).

Supplementary table 1- malpresentation is not the same as obstructed labour

Big baby? Should not be an indication for cs- it is reason for referral-

Why is PROM an indication for a cs?

Why are there no failed VBAC's is it because these hospitals do not have patients with a VBAC- are these women referred to a regional hospital

Pre-eclampsia- are these referred to a higher centre?

Response: We have added a footnote under both tables in the supplementary appendix to explain these findings in relation to indications for caesarean delivery in public hospitals in the Western Cape. We have also added to the methods section the indications for referral of a pregnant woman from a district to a secondary hospital (page 6&7).

Discussion

Can you elaborate on the low cs rates? Particularly the one with 14%? Is this because women are transferred to a higher level? The limitations of not getting individual level data means that we don't know who was referred, what about blood transfusion, etc

Response: We have added some discussion around the caesarean delivery rates and reasons for the low rate at hospital D (page 18). We agree the lack of individual data is a limitation and have acknowledged this in the limitations (page 21).

Page 18-Line 52. I don't think this statement is true- the seniority of the GP's would have to be discussed. These doctors are clearly more senior to the comm serves in these hospitals. It may be that they worked in Obstetrics as MO's before going into private practice.

Response: We have added additional discussion regarding the rate of adverse outcomes as they relate to level of obstetric risk at district hospitals (page 18).

Reviewer: 4**Dr. Stefan Gebhardt, Stellenbosch University**

Comments to the Author:

This is a well written paper and important for any country exploring models for private-public initiatives. The major limitation is acknowledged, and that is that there was no folder review, so the outcomes could very well be different if actual reasons for referral or re-admission were scrutinized. There was one maternal death amongst the 5048 deliveries- was this death related to CS and/or difficulty during CS, or unrelated to any surgery or anesthetics?

Response: we have added a footnote under table 3 to clarify the cause of the maternal death and included this in the discussion (page 19).

Another limitation worth mentioning is that GPs skilled in obstetric practice in rural SA are mostly from an older generation, and the younger/newer doctors have similar lack of skills than their first year public sector medical officer counterparts, as there is no specific GP obstetrics training required. Will this model also work in a setting where the newer generation of GPs are not skilled in safe CS or safe anesthesia?

Response: We agree and have expanded on this point in the discussion (page 20&21).

In smaller SA towns, where GPs also do private obstetrics, these private patients usually deliver in the public sector hospital as it is the only hospital in town. When this 'private' patient develops complications and can no longer afford private care, she is handed over to the public sector (but may still receive a 'free' CS from her GP if he/she is on call that night). This is a potential loophole that can be exploited, as the GP gets paid for private consultations and for the hospital work. Was this option discussed in the interviews?

Response: Yes this question was asked in the interviews. In these five hospitals the private GPs did not have medical indemnity for obstetric practice and only performed caesarean deliveries during their public sector contracted time. This has been clarified on page 7.

VERSION 2 – REVIEW

REVIEWER	Bijlmakers, Leon Radboud University Medical Center, Radboud Institute for Health Sciences (RIHS), Health Evidence
REVIEW RETURNED	20-Dec-2022

GENERAL COMMENTS	I'm satisfied with the revisions and the answers to my earlier suggestions and those of my fellow reviewers. Just one minor last point, pertaining to the last sentence of the very first paragraph (in the Introduction): not all readers will know what 'excess' caesarian deliveries refers to. It would be good to explain that.
--

REVIEWER	Gebhardt, Stefan Stellenbosch University
REVIEW RETURNED	04-Jan-2023

GENERAL COMMENTS	All my concerns were addressed by the revised version.
--

VERSION 2 – AUTHOR RESPONSE

Reviewer: 1

Dr. Leon Bijlmakers, Radboud University Medical Center, Radboud Institute for Health Sciences (RIHS)

Comments to the Author:

I'm satisfied with the revisions and the answers to my earlier suggestions and those of my fellow reviewers. Just one minor last point, pertaining to the last sentence of the very first paragraph (in the Introduction): not all readers will know what 'excess' caesarian deliveries refers to. It would be good to explain that.

Response: we have provide a definition for 'excess' caesarean deliveries on page 4. " The study reported that a 10 per cent increase in voluntary health insurance was associated with a 4 per cent increase in excess caesarean deliveries, defined as caesarean delivery proportions above the global target of 19%.⁴"